# The Nutraceutical Antihypertensive Action of C-Phycocyanin in Chronic Kidney Disease Is Related to the Prevention of Endothelial Dysfunction

**DOI:** 10.3390/nu14071464

**Published:** 2022-03-31

**Authors:** Placido Rojas-Franco, Erick Garcia-Pliego, Alma Gricelda Vite-Aquino, Margarita Franco-Colin, Jose Ivan Serrano-Contreras, Norma Paniagua-Castro, Carlos Angel Gallardo-Casas, Vanessa Blas-Valdivia, Edgar Cano-Europa

**Affiliations:** 1Laboratorio de Metabolismo I, Departamento de Fisiología, Escuela Nacional de Ciencias Biológicas, Instituto Politécnico Nacional, Ciudad de México 07738, Mexico; projasf@ipn.mx (P.R.-F.); erick-gp@hotmail.com (E.G.-P.); mfrancoc@ipn.mx (M.F.-C.); 2Laboratorio de Neurobiología, Departamento de Fisiología, Escuela Nacional de Ciencias Biológicas, Instituto Politécnico Nacional, Ciudad de México 07738, Mexico; griceldagirl@gmail.com; 3Department of Metabolism, Digestion and Reproduction, Division of Systems Medicine, Section of Biomolecular Medicine, Faculty of Medicine, South Kensington Campus, Imperial College London, London SW7 2AZ, UK; j.serrano-contreras@imperial.ac.uk; 4Laboratorio de Farmacología del Desarrollo, Departamento de Fisiología, Escuela Nacional de Ciencias Biológicas, Instituto Politécnico Nacional, Ciudad de México 07738, Mexico; npaniagua@ipn.mx; 5Centro de Investigación en Nutrición y Alimentación Universidad de Chalcatongo Sistema de Universidades Estatales de Oaxaca, Tlaxiaco 71100, Mexico; carlosg84@unicha.edu.mx

**Keywords:** C-phycocyanin, chronic kidney disease, endothelial dysfunction, nephrectomy, antihypertensive

## Abstract

C-phycocyanin (CPC) is an antihypertensive that is not still wholly pharmacologically described. The aim of this study was to evaluate whether CPC counteracts endothelial dysfunction as an antihypertensive mechanism in rats with 5/6 nephrectomy (NFx) as a chronic kidney disease (CKD) model. Twenty-four male Wistar rats were divided into four groups: sham control, sham-treated with CPC (100 mg/Kg/d), NFx, and NFx treated with CPC. Blood pressure was measured each week, and renal function evaluated at the end of the treatment. Afterward, animals were euthanized, and their thoracic aortas were analyzed for endothelium functional test, oxidative stress, and NO production. 5/6 Nephrectomy caused hypertension increasing lipid peroxidation and ROS production, overexpression of inducible nitric oxide synthase (iNOS), reduction in the first-line antioxidant enzymes activities, and reduced-glutathione (GSH) with a down-expression of eNOS. The vasomotor response reduced endothelium-dependent vasodilation in aorta segments exposed to acetylcholine and sodium nitroprusside. However, the treatment with CPC prevented hypertension by reducing oxidative stress, NO system disturbance, and endothelial dysfunction. The CPC treatment did not prevent CKD-caused disturbance in the antioxidant enzymes activities. Therefore, CPC exhibited an antihypertensive activity while avoiding endothelial dysfunction.

## 1. Introduction

Chronic kidney disease (CKD) is defined as the anatomical or functional alteration of the kidney, which is presented by a decrease in the glomerular filtration, albuminuria, or markers of kidney damage (hematuria, urinary sediment abnormality, and structural abnormalities) present for at least three months. CKD is associated with non-communicable diseases such as systemic arterial hypertension (SAH), type 2 diabetes mellitus, and cardiovascular diseases [1]. CKD is a global burden, as it is estimated to cause 864,226 deaths representing 1–5% of mortality worldwide. Low- and middle-income countries are the most affected because their Public Health Systems do not cover all patients’ treatment costs [2,3]. However, cost-effective treatments are available to halt or slow the CKD progression and prevent complications. Late diagnosis and lack of prevention culture also promote CKD complications that must be treated with renal replacement therapy, promoting the socially disadvantaged because the treatment cost is expensively inaccessible for most of the population [4].

5/6 nephrectomy (NFx) is a heuristic model of CKD characterized by a serum creatinine increase, proteinuria, renal remodelation with fibrosis, and hypertension [2,5]. SAH is a symptom in patients with CKD to maintain the glomerular filtration rate by increasing peripheral vascular resistance as an adaptive and compensatory response of the remaining renal mass. Thus, CKD promotes cardiovascular complications related to cardiac and endothelial remodelation, culminating in kidney failure and death [6]. Accordingly, developing new therapies or strategies that slow down disease complications challenges innovation and research.

Vascular endothelium modulates multiple physiological processes, including inflammatory response, angiogenesis, thrombosis, permeability, and vascular tone. Under homeostasis, the endothelium maintains normal blood flow and vascular tone. However, the regular vasoreactivity is altered by chronic insults such as oxidative stress (OS) and inflammatory process that generally are accompanied by vasoconstrictor products and disturbance in hemostasis [7]. Additionally, in humans, there are genetic susceptibilities for endothelial dysfunction in patients with eNOS-encoding gene polymorphisms [8]. In CKD, vascular endothelium damage occurs early and develops complications like SAH, left ventricular hypertrophy, and coronary artery disease [9]. In addition, CKD can induce endothelial and advanced glycation end-products, OS, inflammation, and hoarding of endogenous inhibitors of endothelial nitric oxide synthase (eNOS). These processes reduce NO bioavailability, triggering endothelial dysfunction [10]. The strategies to avoid the NO system production disturbance could delay the endothelial dysfunction as the use of probiotics and nutraceutics that enhance the antioxidant system through the nrf2 signaling pathway [11]. The development of therapies that retard endothelial dysfunction can reduce to CKD-induced cardiovascular complications. *Arthrospira maxima* (Spirulina) and C-phycocyanin (CPC) have been proposed as nutraceuticals in the treatment of acute kidney injury and CKD, which avoids endoplasmic reticulum stress and OS [5,12,13,14]. CPC is a light-absorbing pigment and phycobiliprotein found in cyanobacteria. It is the most-synthesized pigment in *A. maxima* by about 20% of the cell mass. CPC structure comprises a monomer composed of two subunits (α and β), with one chromophore attached to the α subunit and two to the β subunit. The monomers weigh ≈41.4 kDa and come together to form a trimer resembling an (αβ)_3_ ring weighing ≈124.2 kDa. The chromophore, called phycocyanobilin (PCB), is formed by an open-chain tetrapyrrole group that binds to proteins through a thioether bond. PCB gives the molecule its characteristic blue color [15]. Our research group previously reported that CPC treatment prevents SAH, left ventricular hypertrophy, and renal physiological disturbance in the CKD model [5]. However, the complete nutraceutical activity against CKD complications has not been fully elucidated. One hypothesis of CPC nephroprotective action is related to preventing endothelial dysfunction in the CKD, as it was reported that an ethanolic extract of *A. maxima* has a vasomotor response in aorta rings [16]. Therefore, this extract could contain PCB, the active CPC metabolite responsible for scavenging, nephroprotective and anti-inflammatory effects in animal models [17,18,19]. The present study aims to determine whether the antihypertensive activity of CPC is related to the prevention of endothelial dysfunction in the aorta of animals with CKD.

## 2. Materials and Methods

### 2.1. Animals

Twenty-four male Wistar rats weighing 250–280 g were acclimatized in a cool room at a temperature of 21 ± 2 °C, 40–60% relative humidity, and a 12:12 h light/dark cycle with lights on at 8 AM. Food and water were provided ad libitum. The experimental procedures were carried out in accordance with the provisions of the Official Mexican Norm (NOM-062-ZOO-1999 [20], and the approved experimental protocol by the Institutional Bioethics Committee (ZOO-018-2018).

Animals were randomly divided into four groups (*n* = 6): (1) sham + vehicle (phosphate buffer (PB) at pH 7.4 100 mM administered by oral gavage (og), (2) sham + CPC (100 mg/kg/day og), (3) CKD induced by 5/6 Nephrectomy (NFx) + vehicle, (4) NFx + CPC. Animals were habituated to non-invasive blood pressure evaluation two weeks before surgery. 

Animals with a blood pressure ≤ 120/80 mmHg were used for surgical procedure (NFx 5/6), which was performed in anesthetized rats with sodium pentobarbital (16 mg/kg) and xylazine (20 mg/kg) intraperitoneally (ip). During surgery, atropine (2 mg/kg) and caffeine (20 mg/kg) were administered to prevent side effects of anesthesia (ip). Under aseptic conditions, ventral laparotomy was performed and two of the three branches of the left renal artery were occluded with 4-0 black silk sutures (infarction 2/3 of the left kidney), and the right kidney was removed. After surgery, to avoid pain and prevent infections, the animals were administered tramadol (10 mg/kg og) and enrofloxacin (4 mg/kg intramuscular (im)) for two days. CPC treatment started a week after surgery and continued for four weeks. At the end of the treatment, urine samples were collected to evaluate renal function. After euthanization, the thoracic aorta was divided into three sections, two of them were used for functional tests, and the other one was frozen at −70 °C until biochemical and molecular evaluations.

### 2.2. Arthrospira Maxima (Spirulina) and CPC Purification

*A. maxima* was grown in NM medium supplemented with 10% NaHCO_3_ [21] and incubated under conditions previously described [5,13].

The wet cyanobacterial biomass was re-suspended in 30 mL distilled water and broken up with eight freeze-thaw cycles (freezing at −80 °C and thawing at 4 °C). The resultant blend was centrifuged (10 cycles at 21,400× *g* for 10 min at 4 °C) to remove the cell debris. 20 mL of the protean extract was injected into a column (33 cm long × 4.7 cm in diameter) containing Sephadex^®^ G-100 equilibrated with 10 mM of PB (pH 7.4). The bluish fractions obtained were precipitated with a saturated solution of (NH_4_)_2_SO_4_ at 4 °C for 24 h in darkness. This mixture was centrifuged at 21,400× *g* for 2 min at 4 °C, and the resultant pellet was resuspended in 100 mM of PB at pH 7.4 to dialyze in a cellulose membrane with PB for 24 h. CPC was stored at −20 °C with 5 mM of sucrose to await animal administration [21].

A sample of lyophilized CPC was solubilized in PB and subsequently used for the construction of the following calibration curve:
$$CPC\;\left(\frac{mg}{dL}\right)=\frac{\left[Absorbance_{620\;nm}-0.1089\right]}{0.3679};r^{2}=0.9824;r=0.9911;\;\in Absorbance_{620\;nm\;}\left[0.093-2.145\right]$$


The CPC purity was calculated as the A_620_/A_280_ ratio of the maximum absorbance peak of CPC (620 nm) to proteins (280 nm).

### 2.3. Cardiovascular and Renal Functional Evaluations

Systolic and diastolic blood pressure (SBP, DBP), mean arterial pressure (MAP), and heart rate (HR) were determined weekly by a volume pressure recording (VPR), a non-invasive method using the CODA non-invasive instrument (Kent Scientific) coupled to the tail of the rat. 

The day previous to euthanasia, animals were individually placed in metabolic cages (Tecniplast™ Metabolic Cage Systems for Rodents, Thermo Fisher Scientific) to collect 24 h urine. Blood samples were collected before animals were euthanized. Urine protein and creatinine concentration, serum uric acid, and creatinine were measured using Randox kits. The kidney fibrosis was determined by histopathological analysis and the expression of type III collagen (sc-271249, Santa Cruz Biotechnology).

### 2.4. Vasomotor Activity

The thoracic aorta was dissected immediately after euthanizing and then rapidly placed in ice-cold Krebs solution. Each aorta was separated from the surrounding connective tissue and cut transversely into 2 mm long rings, avoiding endothelium damage. Subsequently, they were fixed to the bottom of an isolated organ chamber and to a TSD104 transducer linked to a Biopac (Goleta, CA, USA) to record changes in isometric tension. For this purpose, stainless steel hooks were used. The conditions inside the chamber were: 15 mL of Krebs solution, constant temperature of 37 °C, and continuous carbogen bubbling (mixture of 5% O_2_ and 5% CO_2_). The rings received an initial tension of 2 ± 0.04 g and were allowed to stabilize for an hour. Before starting the experiment, the viability of the aortic tissue of each ring was checked using 0.1 M KCl. Afterward, the tissue was washed three times with Krebs solution to regain the baseline tension. Finally, the aorta rings were contracted with norepinephrine (10^−3^ M), and concentration-response curves of acetylcholine (Ach, 10^−9^ to 10^−5.5^ M) and sodium nitroprusside (SNP, 10^−9^ at 10^−5.5^ M) were performed [22]. In the experiment of endothelium-dependent relaxation to Ach or SNP, the relaxation was expressed as percentage inhibition of the contraction to norepinephrine. Effective concentration 50% (EC_50_) represents the concentration of Ach or SNP that induces fifty percent (50%) inhibition of the contraction to phenylephrine calculated with non-linear regression.

### 2.5. Oxidative Stress and Antioxidant Enzymes Activity in Aorta

Aortas were homogenized in 1 mL of PB (10 mM) to determine oxidative stress markers (lipid peroxidation, reactive oxygen species (ROS) and reduced glutathione (GSH)), the activity of the first-line antioxidant enzymes (catalase, glutathione peroxidase (GPX), glutathione reductase (GR), and superoxide dismutase (SOD) [23].

For lipid peroxidation, 125 μL of the homogenate was mixed with 875 μL of PB. Subsequently, 4 mL of a chloroform-methanol mixture (2:1, *v*/*v*) were added. For phase separation, the tubes were shaken and kept at 4 °C for 30 min (protected from light). After that time, the aqueous phase was aspirated and discarded, and an aliquot of 2 mL of the organic phase was used for fluorescence measurements (370 nm excitation, 430 nm emission). The results were expressed as relative fluorescence units (RFU) per mg of protein.

For ROS quantification, 10 μL of the homogenate were mixed with 1940 μL of TRIS-HEPES mixture (18:1 *v*/*v*), and 50 μL of 2,7-dichlorofluorescine diacetate (DCFH-DA). The tubes were incubated at 37 °C for an hour. The reaction was quenched by freezing, and the fluorescence (488 nm excitation and 430 nm emission) was measured using the presence of 2,7-dichlorofluorescein (DCF). The results were expressed as ng of 2′7′-dichlorofluorescein (DCF) formed/mg protein/h.

An aliquot of 300 μL of the homogenate was treated with 500 μL of 30% phosphoric acid for GSH determination. The mixture was centrifuged at 10,000× *g* for 30 min at 4 °C. 30 μL of the supernatant were diluted in 1.9 mL of FEDTA (100 mM phosphate and 5 mM EDTA). Finally, 100 μL of o-phthaldialdehyde were added to the mixture and measured 10 min later (350 nm excitation and 420 nm emission). The results were expressed as ng of GSH or GSSG/mg protein.

Catalase activity was measured by monitoring the enzyme-catalyzed decomposition of H_2_O_2_. For this, 10 μL of cell extract was mixed with 3 mL of catalase substrate (30 mM H_2_O_2_ in PB). The absorbance was recorded every 30 s for a total time of 90 s at 240 nm and 37 °C. The decomposition of H_2_O_2_ by catalase was determined using its first-order reaction rate constant (k) according to the following equation:k = 2.3t Log [A0/A1]
where: “t” is the time during which the decrease in H_2_O_2_ was measured and A0/A1 is the ratio between the absorbance times (0 and 90 s). Catalase activity is expressed as k/mg of protein.

GPX activity was measured using 2 mL of GPX substrate (0.4 mM EDTA, 10 mM NaN_3_ and 2 mM GSH in) and 60 μL of the cell homogenate. The mixture was pre-incubated for 5 min at 37 °C. After that, 1 mL of 1.25 mM H_2_O_2_ was added to start the reaction. At 5 and 10 min, an aliquot of 500 μL was taken per each and mixed with 1.2 mL of 1.6% metaphosphoric acid. The resultant mixture was diluted (1:2 *v*/*v*) with 400 mM PB, and then put in the presence of 5.5-dinitro-bis-2-nitrobenzoic acid (250 μL). The absorbance was recorded at 420 nm. Results are expressed as μg of GSH used/minute/milligram of protein.

GR activity used 100 µL of a substrate containing 0.2 mM NADPH, 2 mM GSSG in 10 mM Tris-HCl buffer (pH 7). The reaction started when 5 μL of the cell homogenate was added. The absorbance was measured for 10 min (340 nm and 37 °C), determining the consumption of NADPH used by GR to reduce oxidized glutathione (GSSG). Results are expressed as mmol of NADPH consumed/mg protein/ min.

The SOD activity was measured using 2.9 mL of substrate (10 μm NaN_3_, 10 μm reduced cytochrome c, and 1 mM EDTA dissolved in 20 mM NaHCO_3_ and 0.02% triton X-100, pH 10.2). The reaction started adding 50 μL of xanthine oxidase (3.4 mg/mL in 0.1 mM EDTA). The change in absorbance was monitored every 30 s for 3 min at 550 nm. The participation of each SOD isoform was calculated as total activity minus the activity inhibited by 50 µL of 1 mM KCN (selective inhibitor of Cu/Zn-SOD). A unit of SOD activity was defined as that amount of enzyme that decreased the reduction rate of cytochrome c by 50%. One unit of SOD activity was defined as that amount of enzyme that decreased the reduction rate of cytochrome c by 50%.

### 2.6. Nitrite Quantification and Expression of eNOs and iNOS by Western Blot Analysis

Nitrite determination used 500 μL of homogenate in 500 μL of concentrated hydrochloric acid. Then, 500 μL of 20% zinc suspension was added and incubated at 37 °C for an hour. Afterward, the mixture reaction was centrifugated at 4000× *g* for 2 min. 50 μL of the supernatant were placed into a 96-well polystyrene plate with 100 μL of a mixture of 0.6% sulfanilamide and 0.12% N-(naphthyl)-ethylenediamine (1:1 *v*/*v*), the whole mixture was incubated for 15 min at room temperature, and the absorbance was recorded at 530 nm. Nitrite quantification is expressed as μg of NO_2_/mg protein.

eNOS and iNOS expression was determined by Western blot assays using 100 μL of the homogenate mixed with 50 μL of a complete protease inhibitor cocktail (Millipore Sigma, Burlington, MA, USA, Hercules, CA, USA, 161-0737). The samples were homogenized by vortexing and placed in a boiling water bath for 3 min, then kept at −20 °C until being processed. 50 μg of proteins were loaded with sodium dodecyl sulfate (SDS-PAGE) on 10% polyacrylamide gel and separated by electrophoresis (110 V for 60 min). Proteins were then electroblotted from the gel onto a PVDF membrane in a Trans-Blot Turbo system (Biorad) at 25 V and 2.05 A for 7 min. After this period, the membrane was blocked for an hour under constant agitation in PBST (PBS with 0.05% Tween 20 and 5% low-fat milkSvelty^®^). Membranes were incubated overnight at 4 °C in blocking buffer with primary antibodies from Santa Cruz Biotechnology (eNOS sc-376751 and iNOS sc-7271) diluted 1:1000. After incubation, the membranes were washed three times with fresh PBST (20 min/wash) and then incubated in a secondary antibody diluted 1:1500 (goat anti-mouse-HPR from Santa Cruz Biotechnology) under agitation and room temperature for an hour. Then, the membranes were washed three times with fresh PBST. Finally, the protein bands were revealed on photographic plates by chemiluminescence using Luminata TM Forte (MilliporeSigma, Burlington, MA, USA). Expression of the constitutive protein β-actin was used as a positive loading control (Santa Cruz Biotechnology; sc-1615, 1:4000 dilution). The optical density (OD) of all bands was quantified by the Image J program (NIH, Bethesda, MD, USA). Protein OD is expressed as a protein/*β*-actin ratio.

### 2.7. Statistical Analysis

All data are expressed as the mean ± standard error. The PS, PD, MAP, HR were examined by repeated measure (RM) two-way analysis of variance (ANOVA) with time and treatment as factors. Relaxation percentage and maximum relaxation response were root square arcsin transformed. The relaxation responses were examined by RM-two-way ANOVA with concentration and treatment as factors. Meanwhile, the maximum response and the rest of the variables were examined by two-way ANOVA, considering NFx and CPC treatment as factors. ANOVA was followed by the Student–Newman–Keuls (SNK) post hoc test. Statistical significance was considered at *p* < 0.05.

In the experiment of endothelium-dependent relaxation to Ach or SNP, effective concentration 50 % (EC_50_) was calculated with non-linear regression.

## 3. Results

Table 1 shows that the nephrectomized animals developed CKD due to proteinuria and increased uric acid, BUN, and creatinine clearance levels. Furthermore, 100 mg/kg CPC administration partially prevented the increase of these renal function parameters.

Representative photomicrographs of the kidney for all treatments can be observed in Figure 1A, where CPC has an effect on morphological renal remodelation. The sham groups had a normal appearance of the cortex, which contains glomerulus formed by capillaries surrounding the proximal and distal tubules without damage signs. However, NFx caused renal remodeling observed by the onion skinning (i.e., a histological sign of hyperplastic arterioles), fibrilar eosinophilic deposits into the arteriolar media layer, glomerulosclerosis, necrotizing glomerulitis, wall capillaries collapse, and tubular atrophy. According to the grades of chronic changes based on total renal chronicity score, this group had nine points with severe chronic changes (> 50% of glomerulosclerosis, interstitial fibrosis, and tubular atrophy). Furthermore, the type III collagen expression was increased by about six-fold to the sham group (Figure 1C). Meanwhile, CPC treatment in the NFx group prevented CKD-caused renal remodelation, as segmentary and focal glomerulosclerosis (by about 40%) and mild proliferative glomerulosclerosis (by about 10%) were observed. A mild media layer thickened to NFx + vehicle was also observed. This group had moderate chronic change with five points (26–50% of glomerulosclerosis and interstitial fibrosis with 10–25% of tubular atrophy) with moderate chronic changes.

Cardiovascular evaluations are illustrated in Figure 2, where NFx caused an increase of the SBP (A), DBP (B), MAP (C), and HR (D) since the third week after surgical procedure. These elevations are related to hypertension induced by CKD. Meanwhile, the treatment with CPC prevented hypertension during all treatment because SBP, DBP, MAP, and HR values are the same as for the control group.

Figure 3 depicts the evaluation of endothelial dysfunction. Exposure of aorta segments to Ach and SNP resulted in dose-dependent relaxation. However, only the dose-response curves to Ach in aorta rings of nephrectomized rats are shifted rightward (A). The EC_50_ value (−7.03, expressed in log_10_ molar units) for relaxation was greater by approximately three-fold in nephrectomized animals in comparison to the sham group (EC_50_ −6.49, expressed in log_10_ molar units, B). Maximal relaxation with ACh was also decreased in nephrectomized animals compared with the sham group (C). Regarding SNP, only the maximal relaxation was reduced in nephrectomized animals. The treatment with CPC normalized the response of aorta rings to both Ach and SNP.

The evaluation of oxidative stress is shown in Figure 4. Nephrectomized animals showed higher aorta oxidative stress, indicated by the corresponding increase in lipid peroxidation (≈150%, A) and ROS (≈205%, B). The NFx also reduced the GSH content (≈47%, C). On the other hand, CPC prevented the increase of lipid peroxidation, ROS while ameliorating the reduction in GSH content.

Regarding the first-line enzymatic antioxidant system (Figure 5), NFx decreased the activity of catalase (≈50%, A), GR (≈58%, C), total SOD (≈30%, D), and Cu/Zn-SOD (≈50%, F) without changes in the GPX and Mn-SOD activities (B and E, respectively). However, CPC treatment in nephrectomized animals did not ameliorate the antioxidant enzymatic system.

The effect of CPC on NFx-induced disturbance in NO production is illustrated in Figure 6. Nephrectomized rats decreased nitrite content (~30%, A) due to a down-expression of eNOS (~41%, B). These animals also over-expressed iNOS (~450%, C). CPC partially reduced the nitrite dysfunction by ~30% and ~10% for nitrites reduction and eNOS down-expression, respectively. In addition, iNOS expression was normalized in nephrectomized animals treated with CPC.

## 4. Discussion

This study demonstrated that CPC has a nutraceutical effect on the kidney, preventing renal fibrosis in the remanent kidney, SAH, and cardiovascular complications [12]. Although the antihypertensive effect of CPC was previously reported [5,24], this study proposed that pharmacological nutraceutical mechanisms are associated with the effect of CPC on vascular endothelium.

CPC prevented endothelium dysfunction by reducing CKD-caused oxidative stress and NO production disturbance. The explanation of this outcome involved the metabolism of CPC to chromo-peptides such as PCB and other vasoactive peptides. CPC is a scavenger, and this is the first mechanism against CKD-caused endothelial dysfunction. Previous reports proposed that CPC acts as a prodrug because it is a protein with a molecular weight of about 120 KDa. When it is administrated by oral gavage, it suffers breakdown by the mammal’s metabolism, and probably also by the gut microbiota, releasing active compounds and promoting therapeutical effect. The CPC is metabolized by gastric acid in the stomach and intestinal peptidases in the digestive tract. CPC is broken up into peptides, chromo-peptides, and PCB, which are absorbed. In the case of the PCB, it must be transported by albumin into the plasma to exert its action on target cells [25,26]. CPC through PCB has nephroprotective action, as previously reported [17,27]. PCB is a linear tetrapyrrole that acts as a potent nucleophilic compound similar to bilirubin and biliverdin that neutralizes free radicals, ROS, and reactive nitrogen species, normalizing the redox environment disturbance [17]. In CKD, oxidative stress triggers endothelial dysfunction mediated by the increase of NADPH oxidase complex activity [10]. However, PCB metabolized by the biliverdin reductase pathway produces phycocyanorubin, a potent inhibitor of NADPH oxidase complexes [28]. PCB also reduces the expression of its oxidant components (Nox4, p22phox, and p47phox), causing inhibition of NADPH-induced superoxide production [27].

Nevertheless, the CPC treatment prevented high levels of lipid peroxidation and ROS and low levels of GSH. The activity of antioxidant enzymes such as catalase, GR, and Cu/Zn-SOD disturbance was not avoided. This indicates that CPC partially prevented oxidative stress from delaying endothelial dysfunction.

Moreover, chronic oxidative stress in CKD promotes inflammation and fibrosis in the remanent functional kidney, potentiating endothelial dysfunction [9]. However, CPC was reported to function as an anti-inflammatory nutraceutical because it reduces the expression of COX2 and NOS2 [29]. 

The anti-inflammatory activity of CPC can be related to PCB, as it is able to reduce gene expression associated with the proinflammatory environment, such as IFN-γ, CCL12, IL-4, CD74, Foxp3, TGF-β 1CXCL2, ICAM-1, IL-17A, C/EBPβ, IL-6, TNF-α, and IL-1β [30]. Thus, CPC treatment prevented CKD-caused chronic oxidative stress, avoiding redox environment alteration in aorta rings and delaying the endothelial dysfunction. Additionally, concerning renal fibrosis, the CPC anti-inflammatory action prevents myofibroblast activation, reducing the expression of connective tissue growth factor and α-smooth muscle actin. Thus, the CPC antinflamatory and antifibrotic activities avoid the constant reduction in the remanent renal function in CKD [31]. 

CPC could also be an additional antihypertensive mechanism independent of the antioxidant and anti-inflammatory action, which contributes to delaying cardiovascular complications. This assumption relies on other studies reported that employed potent antioxidants such as S-allylcysteine or curcumin in the same CKD model, where the SAH was partially prevented, although they reduced chronic oxidative stress [32,33]. Therefore, the possible mechanism involves the peptides produced from the CPC metabolism in the digestive tract. This hypothesis relies on the fact that, in simulated human gastrointestinal digestion of Spirulina (with about 10% of CPC), tripeptides and decapeptides are produced with vasoactive action through the PI3K/AKT/eNOS pathway. In addition, one of the peptides (YNKFPY) with mild vasorelaxant activity from hydrolysis of Spirulina was associated with the α-chain [34,35,36]. Therefore, this hypothesis must be further studied using the simulated human gastrointestinal digestion of CPC to prove the antihypertensive effect of its hydrolyzed peptides. 

We also observed that CPC prevented the GSH reduction caused by CKD in aorta rings, avoiding S-glutathionylation of eNOS and other proteins which participate in endothelial dysfunction [34]. CPC also prevented the down-expression of eNOS and overexpression of iNOS in CKD animals. Although the prevention of NO production was not observed, it was possible to detect that NO formation was partially disturbed in animals with CKD treated with CPC. CKD causes endothelial dysfunction denoted by the Ach dose-effect curve that shifts rightwards, and the EC_50_ is three-fold higher than the sham group. Additionally, the maximum vasodilatation due to SNP is lower than the sham group. The CPC treatment partially prevented endothelial dysfunction. The endothelium is the central regulator of vascular homeostasis that maintains the balance between vasodilation and vasoconstriction by producing several molecules such as NO, which play an essential role in the regulation process. Upsetting this balance due to oxidative stress and inflammation leads to endothelial dysfunction damaging the arterial wall. In CKD, endothelial dysfunction is responsible for cardiovascular complications such as SAH and left ventricular hypertrophy [10]. CPC is a scavenger of ROS, and its metabolite PCB neutralizes peroxynitrite, a toxic molecule that participates in endothelial dysfunction [19]. This antioxidant CPC action avoids CKD-caused endothelial damage and cardiovascular complication such as SAH. 

Based on our result about the antihypertensive action of CPC and other nutraceuticals, such as anti-inflammatory, antioxidant, and scavenger, this protein has the potential to treat several diseases related to endothelial dysfunction. In the case of CKD, CPC could delay the development of cardiovascular complications with a positive role in ameliorating the quality of life. 

Although CPC had an anti-inflammatory activity, the aorta has not yet been evaluated in the model of CKD. It is the first limitation of this study, but it could explain the complete mechanism of endothelial dysfunction. CPC could be a prodrug that must be metabolized to induce therapeutical effects. However, this hypothesis is not fully described and must be further evaluated to more deeply understand the pharmacological mechanism of its antihypertensive activity.

## 5. Conclusions

This study demonstrates that the antihypertensive action of C-phycocyanin (CPC) is related to avoiding endothelium dysfunction in a model of CKD. The mechanism that explains the nutraceutical effect is associated with reducing oxidative stress, iNOS downregulation, the increase of GSH, and the over-expression of eNOS in aorta rings. In addition, CPC also delayed CKD progression by preventing renal fibrosis, SAH, and endothelial dysfunction. These nutraceutical properties exerted by CPC make it an excellent candidate for clinical trials to delay CKD progression.

## Figures and Tables

**Figure 1 nutrients-14-01464-f001:**
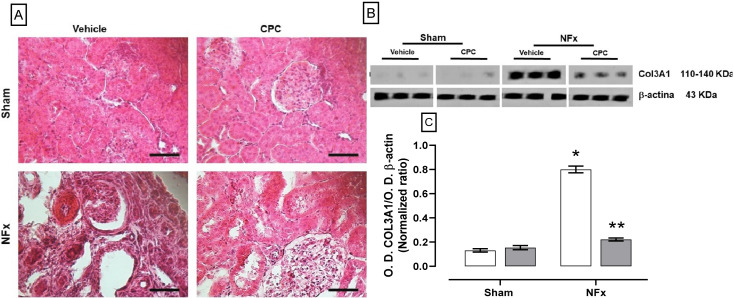
Effects on renal tissue for the four experimental groups. (**A**) Representative photomicrographs of the renal cortex stained by hematoxylin & eosin. The lower right bar of photomicrographs represents 250 μm. (**B**) Blot of the COL3A1 expression. (**C**) Type III collagen expression of NFx rats. (*) *p* < 0.05 compared to the sham + vehicle group. (**) *p* < 0.05 compared to the NFx + vehicle group. Two-way ANOVA and SNK *post hoc* test.

**Figure 2 nutrients-14-01464-f002:**
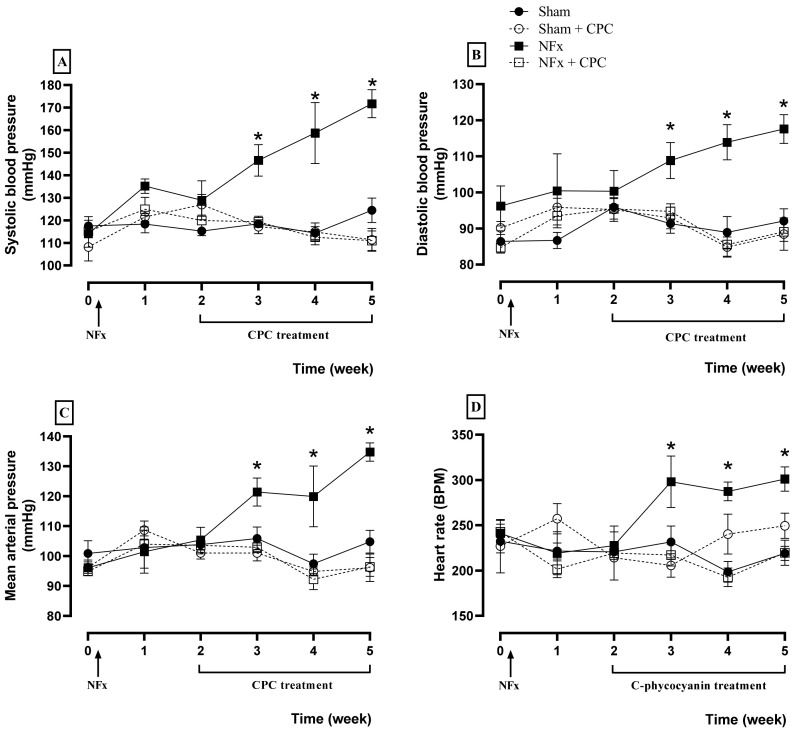
Effect of C-phycocyanin on systolic blood pressure (**A**), diastolic blood pressure (**B**), mean arterial pressure (**C**), and heart rate (**D**) of NFx rats. Values represent the mean ± SEM. (*) *p* < 0.05 compared to the sham group at the same time. RM two-way ANOVA and SNK post hoc test.

**Figure 3 nutrients-14-01464-f003:**
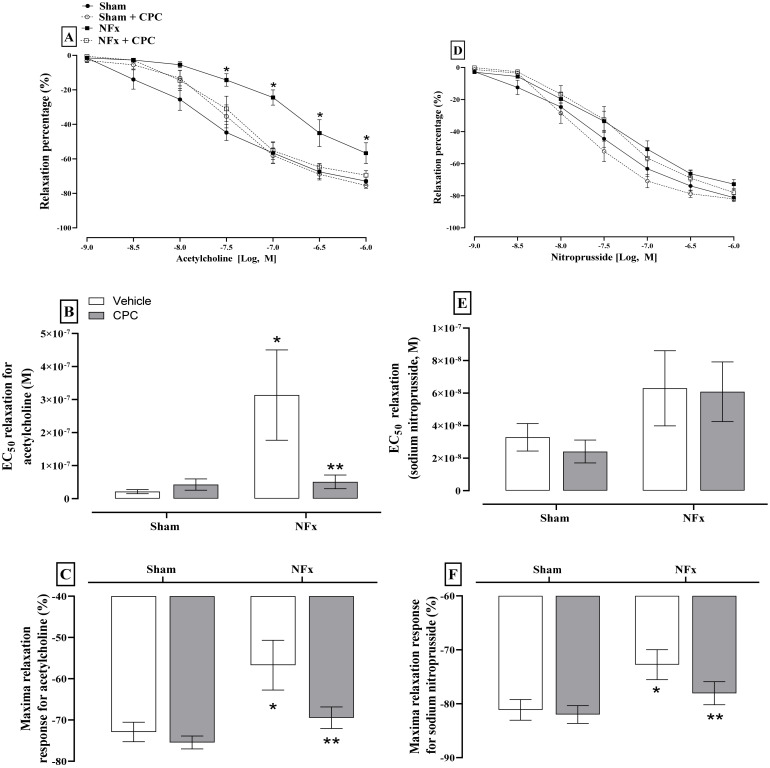
Effect of C-phycocyanin on endothelium and muscular functions of aorta rings of 5/6 nephrectomized rats. (**A**–**C**) show the acetylcholine relaxation response, and (**D**,**E**) show the sodium nitroprusside relaxation response. Values represent the mean ± SEM. In (**A**,**D**) (*) *p* < 0.05 compared to the sham + vehicle at the same concentration. RM Two-way ANOVA and SNK post hoc test. In (**B**,**C**,**E**,**F**) (*) *p* < 0.05 compared to the sham + vehicle. (**) *p* < 0.05 compared to the NFx + vehicle. Two-way ANOVA and SNK post hoc test.

**Figure 4 nutrients-14-01464-f004:**
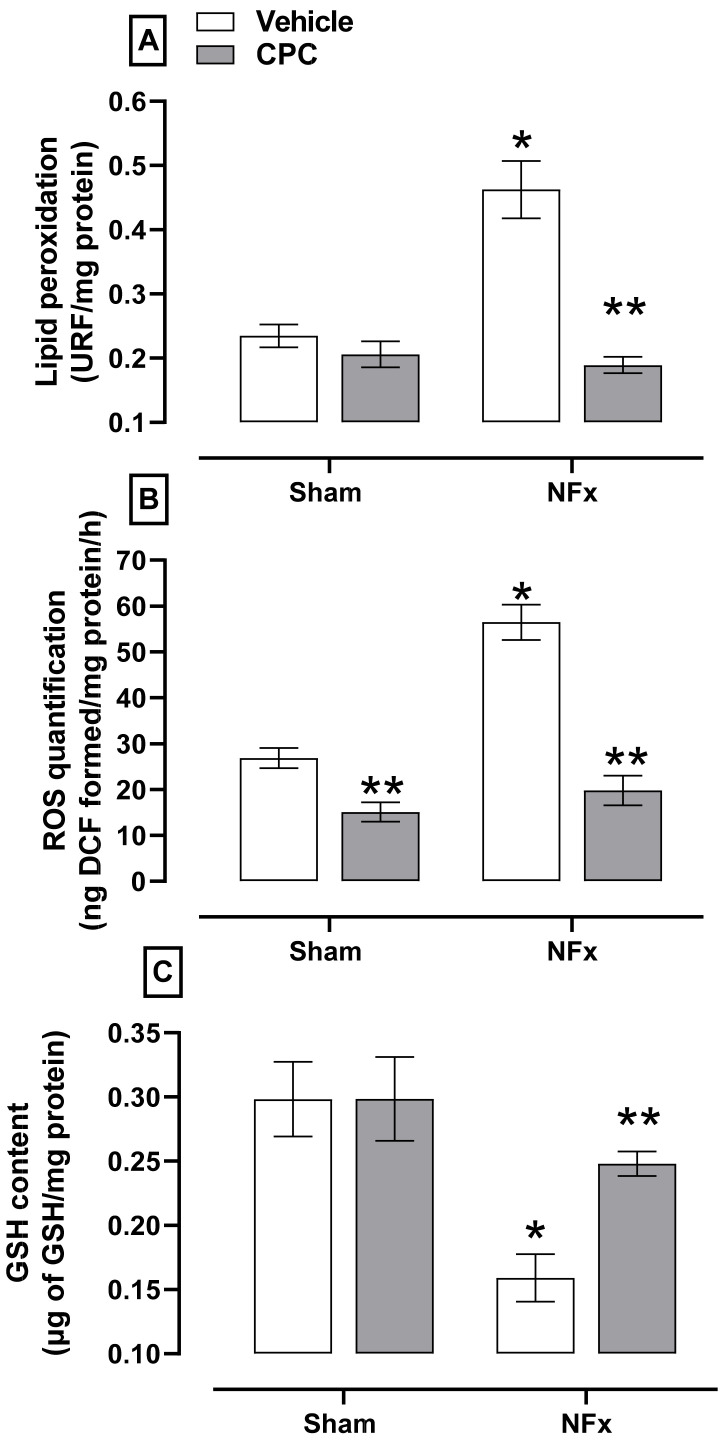
Effect of C-phycocyanin on oxidative stress markers. (**A**) Lipid peroxidation, (**B**) ROS and (**C**) GSH as redox environment marker in the aorta of **NFx** rats. Values represent the mean ± SEM. (*) *p* < 0.05 compared to the sham + vehicle at the same concentration. (**) *p* < 0.05 compared to the NFx + vehicle. Two-way ANOVA and SNK post hoc test.

**Figure 5 nutrients-14-01464-f005:**
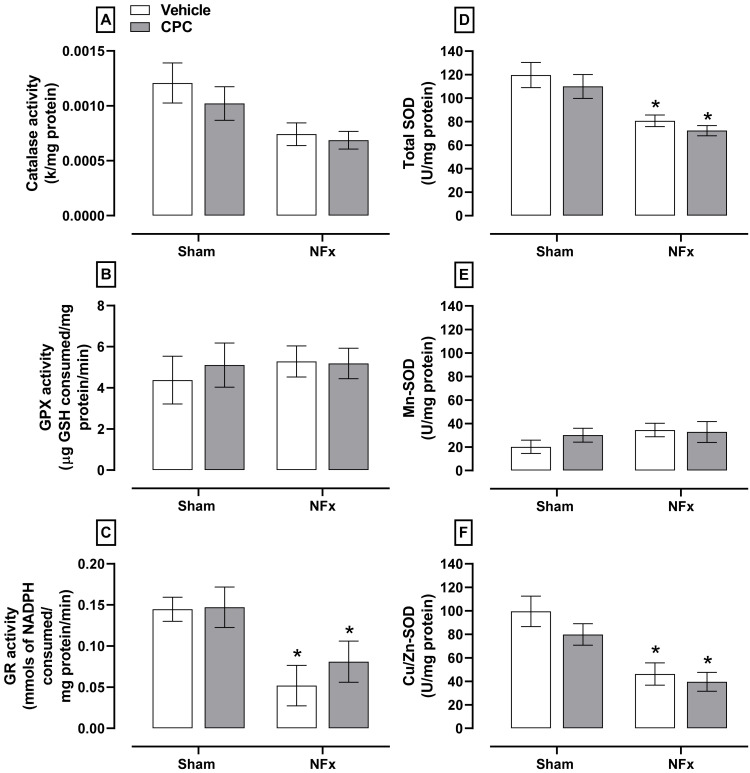
Effect of CPC on the activity of first-line antioxidant enzymes in aorta such as catalase (**A**), GPX (**B**), GR (**C**), total SOD (**D**), Mn-SOD (**E**) and Cu/Zn-SOD (**F**). Values represent the mean ± SEM. (*) *p* < 0.05 compared to the sham + vehicle at the same concentration. Two-way ANOVA and SNK post hoc test.

**Figure 6 nutrients-14-01464-f006:**
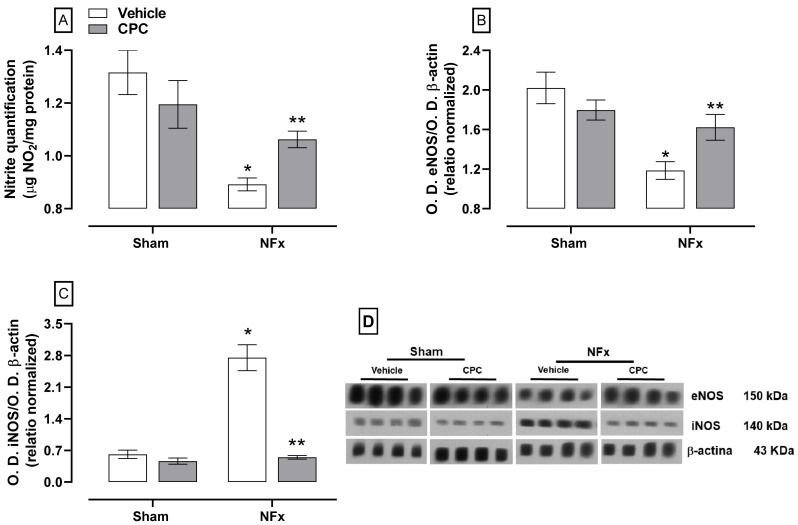
Effect of CPC on nitrite quantification (**A**), eNOS (**B**) and iNOS (**C**) expression in the aorta. (**D**) Blot of the protein expression. Values represent the mean ± SEM. (*) *p* < 0.05 compared to the sham + vehicle at the same concentration. (**) *p* < 0.05 compared to the NFx + vehicle. Two-way ANOVA and SNK post hoc test.

**Table 1 nutrients-14-01464-t001:** Effect of 100 mg/kg CPC administration on renal function in NFx rats.

	Sham	NFx	Sham + CPC	NFx + CPC
**Proteinuria ^a^**	3.59 ± 0.94	10.13 ± 0.99 *	3.12 ± 0.58	6.4 ± 0.58 *,**
**Uric acid ^a^**	3.48 ± 0.75	5.44 ± 0.77 *	3.27 ± 0.97	3.56 ± 0.65 *
**BUN ^a^**	62.91 ± 4.04	86.06 ± 11.48 *	58.09 ± 8.64	72.02 ± 3.9 *,**
**Creatinine clearance ^b^**	0.57 ± 0.1	1.32 ± 0.07 *	0.55 ± 0.17	0.85 ± 0.11 *,**

Data represent mean ± standard error. (*) *p* < 0.05 compared to the sham + vehicle. (**) *p* < 0.05 compared to the NFx + vehicle. Two-way ANOVA and SNK post hoc test. ^a^ in mg/dL. ^b^ in mL/min.

## Data Availability

The data sets collected and analyzed in the present study can be found here [https://drive.google.com/drive/folders/1iuYMRVw_fspUNeFjjIwxoXPtC1mHINwK?usp=sharing] since 30 March 2022.

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
