# Peer review of "The Nutraceutical Antihypertensive Action of C-Phycocyanin in Chronic Kidney Disease Is Related to the Prevention of Endothelial Dysfunction"

_nutrients, 2022, doi:10.3390/nu14071464_

Round 1

Reviewer 1 Report

Rojas-Franco et al showed that TheAntihypertensive Action of C-phycocyanin 3 in Chronic Kidney Disease is Related to the Prevention of Endothelial Dysfunction. However some concerns should be adressed.

Kidney Fibrosis is a hallmark of CKD, histological slides should be included , as pathological images is golden standard for CKD

Author Response

March 11th , 2021

Editor-in-Chief.

Nutrients

We complete the first revision of the article number “nutrients-1606871”.

We appreciate your time for the article revision. Your comments helped us very much for an amending version. We attended all the points that the reviewers questioned us and the English language and style was improved

Reviewer 1

Kidney Fibrosis is a hallmark of CKD, histological slides should be included, as pathological images is golden standard for CKD

Response: We processed kidneys for histopathological study and we evaluated the type III collagen by western blot. Thus it was added the figure 1 to resolve this point.

Reviewer 2:

The manuscript focuses on the possible role of C-phycocyanin (CPC) in contrasting endothelial dysfunction and hypertension in  CKD models. The manuscript is well written. However I would suggest some minor revision to improve your manuscript:

  • Genetic predisposition to endothelial dysfunction is an emerging and crucial aspect involved in several systemic diseases, such as cardiovascular diseases (see J Cardiovasc Dev Dis. 2021 Sep 18;8(9):116. doi: 10.3390/jcdd8090116.). It is important in prevention strategies and in the identification of therapeutic options. Please briefly discuss the role of genetic predisposition as a further determinant of endothelial dysfunction and the role of nutraceuticals in regulating NO and eNOS expression (see Am J Physiol Heart Circ Physiol. 2021 Nov 1;321(5):H839-H849. doi: 10.1152/ajpheart.00278.2021.)

Response: We added the cites because they improve the central idea of the endothelial dysfunction into introduction.

  • please define all the abbreviation first time they appear in the text, figures, tables and abstract

Response: We amended it.

  • please add a brief discussion regarding possible future implication of the results of your research for human medicine.

Response: We added a possible implication of the CPC treatment in cardiovascular disease.

Kind regards

Sincerely.

Edgar Cano-Europa and Vanessa Blas-Valdivia, Ph. D.

Laboratorio de Metabolismo I, Departamento de Fisiología, ENCB. Instituto Politécnico Nacional. Avenida Wilfrido Massieu sin número, esquina Manuel Estampa, Colonia Unidad Profesional Adolfo López Mateos, Delegación Gustavo A. Madero, Código Postal 07738, Ciudad de México, México. Fax (525)729-62-06 email [email protected], [email protected].

Reviewer 2 Report

The manuscript focuses on the possible role of C-phycocyanin (CPC) in contrasting endothelial dysfunction and hyepertension in  CKD models. The manuscritp is well written. However i would suggest some minor revision to improve your manuscript:

1) Genetic predisposition to endothelial dysfunction is an emerging and crucial aspect involved in several systemic diseases, such as cardiovascular diseases (see J Cardiovasc Dev Dis. 2021 Sep 18;8(9):116. doi: 10.3390/jcdd8090116.). It is important in prevention strategies and in the identification of therapeutic options. Please briefly discuss the role of genetic predisposition as a further determinant of endothelial dysfunction and the role of nutraceuticals in regulating NO and eNOS expression (see Am J Physiol Heart Circ Physiol. 2021 Nov 1;321(5):H839-H849. doi: 10.1152/ajpheart.00278.2021.)

2) please define all the abbreviation first time they appear in the text, figures, tables and abstract

3) please add a brief discussion regarding possible future implication of the results of your research for human medicine.  

Author Response

(The authors gave the same response as above.)
